# Complex Transposon Insertion as a Novel Cause of Pompe Disease

**DOI:** 10.3390/ijms221910887

**Published:** 2021-10-08

**Authors:** Igor Bychkov, Galina Baydakova, Alexandra Filatova, Ochir Migiaev, Andrey Marakhonov, Nataliya Pechatnikova, Ekaterina Pomerantseva, Fedor Konovalov, Maria Ampleeva, Vladimir Kaimonov, Mikhail Skoblov, Ekaterina Zakharova

**Affiliations:** 1Research Centre for Medical Genetics, 115478 Moscow, Russia; gb2003@yandex.ru (G.B.); maacc@yandex.ru (A.F.); migyaev@yandex.ru (O.M.); marakhonov@generesearch.ru (A.M.); mskoblov@gmail.com (M.S.); doctor.zakharova@gmail.com (E.Z.); 2Morozov Children’s City Clinical Hospital, 119049 Moscow, Russia; Funny85@bk.ru; 3Center of Genetics and Reproductive Medicine GENETICO, JSC, 119333 Moscow, Russia; e.pomerantseva@gmail.com (E.P.); kaimonov@genetico.ru (V.K.); 4Independent Clinical Bioinformatics Laboratory, 123181 Moscow, Russia; fk@clinbio.ru (F.K.); ampleeva@clinbio.ru (M.A.)

**Keywords:** SVA, L1, transposable elements, transcription termination, missplicing, retrotransposon, transposon insertion, lysosomal storage disease, functional analysis

## Abstract

Pompe disease (OMIM#232300) is an autosomal recessive lysosomal storage disorder caused by mutations in the *GAA* gene. According to public mutation databases, more than 679 pathogenic variants have been described in *GAA*, none of which are associated with mobile genetic elements. In this article, we report a novel molecular genetic cause of Pompe disease, which could be hardly detected using routine molecular genetic analysis. Whole genome sequencing followed by comprehensive functional analysis allowed us to discover and characterize a complex mobile genetic element insertion deep in the intron 15 of the *GAA* gene in a patient with infantile onset Pompe disease.

## 1. Introduction

Pompe disease (PD, OMIM#232300) is an autosomal recessive lysosomal storage disease with an average estimated incidence of 1 in 13,000 [1]. PD is caused by a deficiency of acid alpha-glucosidase (GAA), a lysosomal enzyme essential for glycogen degradation and encoded by the *GAA* gene [2]. The excess accumulation of glycogen in lysosomes leads to their progressive enlargement and rupture [3]. The dysfunction of lysosomes leads to defective autophagy, calcium homeostasis, oxidative stress and mitochondrial abnormalities [4]. The most affected tissues are skeletal, cardiac and smooth muscles and nervous system.

Depending on the severity of genetic defect and residual GAA activity PD can manifest at any age [5]. The classic infantile-onset form is characterized by the age of onset at ≤12 months with failure to thrive, cardiomyopathy, hypotonia, muscle weakness and respiratory insufficiency [6,7] (Updated on 11 May 2017). The gold standard for the diagnosis of PD is the blood-based mass spectrometry assay for measurement of GAA activity, followed by the molecular genetic analysis of the *GAA* gene [5,8]. The fast and accurate diagnosis of PD is important due to the presence of pseudo-deficiency alleles and the availability of enzyme replacement therapy.

The *GAA* gene is about 28 kb long, located at chromosome 17q25.2-q25.3 and contains 20 exons, which encode a protein of 952 amino acids. To date, more than 679 mutations are described in *GAA* according to HGMD (http://www.hgmd.cf.ac.uk) (revised on 15 July 2021), most of which are missense, small deletions and splicing variants distributed evenly throughout the gene [9]. Also, several pathogenic variants in promoter region and 5′ UTR, which alter *GAA*s expression were reported [10]. No strict genotype–phenotype correlations were established for PD. The exception is a frequent mild c-32-13T>G pathogenic variant, which is mainly associated with childhood/adult phenotypes and seems to prevent the occurrence of a severe infantile-onset form of PD [11].

In this article, we report the novel type of *GAA* mutation, which represents the transposable element (TE) insertion deep in the intron 15 of the gene. TEs constitute the significant part of eukaryotic genomes and characterized by their ability to move from one DNA loci to another [12,13]. TEs provided to the human genome the large variety of functionally significant sequences but, on the other side, contribute greatly to human genetic diseases. According to the transposition mechanism, TEs are divided into two major classes—RNA transposons or retrotransposons and DNA transposons, which are further divided into subfamilies. The ability to autonomously translocate itself throughout a genome distinguishes autonomous TEs, which encode the proteins such as DNA transposase or reverse transcriptase/integrase, and non-autonomous TEs, which borrow the above proteins for transposition. *Alu*, L1, and SVA elements are the most active retrotransposons in humans and are recently associated with more than 124 cases of genetic disease [14].

## 2. Materials and Methods

### 2.1. Biochemical Analysis

The activity of patient’s alpha-1,4-glucosidase was measured in dried blood spots with NeoLSD™ MSMS kit by ESI-MS/MS using AB SCIEX 3200 Qtrap as described previously [15].

### 2.2. DNA Analysis

The sequences of primers and TE amplicon consensus sequence are available in the Appendix A.

Genomic DNA was extracted from whole blood with EDTA and DBS using GeneJET Genomic DNA Purification Kit (Thermo Fisher Scientific, Waltham, MA, USA). Sanger sequencing was performed on ABI PRISM 3500xL Genetic Analyzer (Thermo Fisher Scientific, Waltham, MA, USA). Whole genome sequencing of the patient’s DNA was performed with TruSeq DNA PCR-Free sample preparation kit on NovaSeq 6000 (Illumina, San Diego, CA, USA). NGS sequencing of the TE amplicon was performed with AmpliSeq technology on Ion S5 (Thermo Fisher Scientific, Waltham, MA, USA). Variants were named according to the *GAA* reference sequence NM_000152.5 and GRCh38.p12 genome assembly.

### 2.3. Bisulfite Sequencing

Bisulfite conversion of DNA was performed with EpiTect Bisulfite Kit (Qiagen, Germantown, MD, Germany). Four primer pairs were designed to amplify the converted DNA: for CpG island in the exon 1 of *GAA*, CpG island spanning exons 4 and 5 and two regions in the 5′ and 3′ end of intron 15. The PCR products were analyzed by Sanger sequencing.

### 2.4. RNA Analysis

Total RNA was isolated from whole blood cells using Leukocyte RNA Purification Plus Kit (Norgene, Thorold, ON, Canada). The first strand of cDNA was synthesized using ImProm-II™ Reverse Transcriptase (Promega, Madison, WI, USA) and oligo(dT) primers. The overlapping fragments of the *GAA* cDNA were amplified by PCR and Sanger sequenced.

### 2.5. Rapid Amplification of cDNA Ends

The rapid amplification of cDNA ends (RACE) was performed after double stranded cDNA synthesis with Mint RACE cDNA amplification set (Evrogen, Moscow, Russia). To establish the 3′ end of the patient’s *GAA* mRNA, three rounds of subtractive hybridization were performed using Step-Out RACE technique [16], Mint RACE primer set (Evrogen, Moscow, Russia) and three gene-specific primers. The obtained PCR product was analyzed by Sanger sequencing.

### 2.6. Real-Time PCR

Real-time quantitative PCR was performed on QuantStudio 5 thermocycler with SybrGreen chemistry. The ddCq method was used for the data analysis [17]. Primers were designed to provide almost 100% reaction efficiency and high specificity (The results of melt curve and standard curve analyzes are demonstrated in Appendix A). The patient’s *GAA* expression was measured with primers spanning exons 3–4 and exons 15–16 of the gene. The *PSMA1*, *EIF3M*, *C11orf58* and *EMC4* housekeeping genes were used as the reference genes.

To estimate the ratio of different parts of the chimeric transcript, the plasmid construct based on pAL2-T vector (Evrogen, Moscow, Russia) was made. Three regions were amplified with tailed primers from the patient’s cDNA: spanning exons 14–15 of *GAA*, spanning the exon 15 and the 5′ part of TP and the one located at the 3′ part of TP. The reference target spanning exons 14–15 was chosen to be as close as possible to the *GAA*-TE junction to minimize the 3′ bias in cDNA molecules, caused by using the oligo(dT) primers for cDNA synthesis. Amplicons were merged together by overlap-extension PCR and cloned into the pAL2-T vector. Plasmids were diluted to the concentration which gives the similar Cq as the studied samples and were used as reference sample for ddCq calculation.

## 3. Results

### 3.1. Patient’s Summary

The patient is a 1.5-month-old male from the fifth birth in a consanguineous marriage. The family also had one miscarriage at 7 weeks, two children with hypertrophic cardiomyopathy, who died before 1 year, and one healthy 9-year-old child (Figure 1A). Medical examination of the patient revealed muffled heart sounds, suspected hypertrophic cardiomyopathy on echocardiography, elevated liver aminotransferases and creatine kinase activity and reduced tendon reflexes in the lower limbs. Based on the anamnesis and the family history, PD was suspected. The subsequent biochemical analysis revealed the decreased activity of alpha-1,4-glucosidase—0.130 μmol/L/h (normal range 1–20), which is the marker of PD.

### 3.2. Molecular Genetic Analysis

To establish the molecular genetic diagnosis of PD, we analyzed the exons with about 200 b.p. of adjacent introns and the promoter region of the patient’s *GAA* gene by Sanger sequencing but did not reveal any rare suspicious variants.

To identify the possible splicing alterations, which could be caused by the deep intronic variants, the RNA extracted from the patient’s white blood cells was analyzed. After cDNA synthesis, the whole coding sequence of the *GAA* mRNA was amplified by overlapping fragments, visualized by agarose gel electrophoresis and Sanger sequenced. No additional PCR products were detected compared to the control sample and Sanger sequencing also did not reveal any abnormalities.

The other considered causes of the altered *GAA* function were the mutations located in regulatory sequences, disruption of topologically associated domains or translocation of the *GAA* locus in the region of condense heterochromatin, that lead to severe reduction of gene’s expression. To quantitatively analyze the *GAA* expression, qPCR primers targeting the 5′ region (3–4 exons) and the 3′ region (exons 15–16) of the *GAA* cDNA were designed. The results of the qPCR revealed that the patient’s *GAA* expression is reduced to 15% of the control at the 5′ region and to 1% at the 3′ region (Figure 1C).

According to the GIS ChIA-PET track of the UCSC genome browser (https://genome.ucsc.edu/) (revised on 15 July 2021), the *GAA* promoter has a strong three-dimensional interaction with the *EIF4A3* promoter and is probably common for these two genes. Assuming that the reduced activity of the *GAA* promoter could affect the expression of *EIF4A3*, we measured its expression by qPCR and revealed that it did not differ from the control. This observation suggested that the causative mutation most probably is located in the *GAA* gene’s body.

At this step the patient’s cDNA was used up and no more mRNA or fresh blood were available because the patient died. Therefore, the mother’s blood was obtained, and all subsequent mRNA analyses were carried out on the mother’s sample.

As the mother can be the carrier of the allele with reduced expression, we performed the analysis of the allelic imbalance at the *GAA*’s mRNA level. The sequencing of the *GAA* exons amplified from the mother’s DNA revealed five heterozygous variants. The sequencing of the cDNA fragments containing these variants demonstrated the characteristic allelic imbalance, similar to the expression pattern in patient’s cDNA—partial loss of zygosity for two variants in the 5′ half of the gene and complete loss of zygosity for three variants in the 3′ half (Figure 1D).

At the next step, whole genome sequencing was performed using the patient’s DNA. Automatic annotation of the genome using the standard variant calling protocol did not reveal any suspicious variants in the *GAA* gene, but the visual inspection of the alignment file identified the duplication of 15 b.p. (NC_000017.11:g.80,114,172_80,114,186dup) within the intron 15 of *GAA*. In addition, the discordant reads flanking this duplication were identified, mapping at a chromosome 20 region that contains polymorphic SVA element named CPX-1 (NC_000020.11:2,822,517-2,825,763) [18] (Figure 2A). SVA is a mobile genetic element, which transcribes from DNA locus and uses the RNA molecule intermediate during transposition. Being nonautonomous, SVA requires L1 proteins for integrating into the genome by target-primed reverse transcription with the formation of target site duplication near the integration site “attaaaa”, which is located in the *GAA* intron 15 [19,20]. The intron 15 of *GAA*, in turn, does not contain any polymorphic transposable elements according to the database of retrotransposon insertion polymorphisms dbRIP (http://dbrip.brocku.ca/searchRIP.html) (revised on 15 July 2021).

Thus, the retrotransposition of the SVA element from chromosome 20 was suspected. The Sanger sequencing of the insertion boundaries established the precise coordinates of the probable retrotransposition origin at NC_000020.11:2,823,027-2,826,302. According to the RepeatMasker track of the UCSC browser, this region contains TEs of the SVA_D, SVA_E and L1ME3 classes (Figure 2B). The complete 3394 b.p. insertion was also amplified from the patient’s DNA with the primers located in the introns of *GAA* and was sequenced by NGS. The obtained sequence matches the reference by 98.95% (Appendix A).

### 3.3. Study of Mechanism of the TE Insertion Molecular Pathogenesis

As the *GAA* intron 15 already contains TE of various types, the pathogenic effect of the novel TE insertion was not obvious. One of the mechanisms by which the cell suppresses the TE activity in its genome is the methylation of the corresponding locus. To identify whether changes in the methylation of the *GAA* gene are the cause of reduced expression, we performed bisulfite sequencing of two CpG islands located near the exon 1, exons 4–5, and two regions located in intron 15. All of the analyzed CpG pairs in patient’s DNA around the exon 1 were unmethylated, and CpG pairs in three other regions were methylated, which was also observed in the control DNA and represents the normal methylation pattern of the genes.

The vast majority of pathogenic TE insertions, reported as a cause of human diseases, are located in exons, alter splicing sites or serve as a source of cryptic splice sites, causing exonization [14]. As the identified SVA/L1 element is located in the forward orientation relatively to the gene and no exonization or other events were detected during amplification of patient’s cDNA fragment spanning exons 15–16, we hypothesize, that the TE sequence could be spliced with the *GAA* mRNA as the last exon and terminate the transcription. Therefore, we performed PCR using the mother’s cDNA with primers located in *GAA* exon 15 and in the 5′ region of transposon (Figure 2C dark grey arrows). This region consists of *Alu* element in the reverse orientation with several splicing acceptor sites of medium strength (according to https://www.fruitfly.org/seq_tools/splice.html (accessed on 15 July 2021)). Sanger sequencing of the PCR products revealed that the *GAA* exon 15 was spliced with TE due to activation of the acceptor site composed of the AG dinucleotide located at the end of CCCTCT repeats, serving as the polypyrimidine tract and the branch point located in the *GAA* intron. It is worth noting that this AG dinucleotide was formed due to the indel, corresponding to NC_000020.11:2,823,108-2,823,110delGATinsCAA, which is absent in the reference sequence.

To determine the 3′ end of the chimeric mRNA isoform we performed the rapid amplification of the cDNA ends (3′ RACE) using primers located in L1 and SVA_E regions of TE (Figure 2C light grey arrows). Sanger sequencing of the PCR products revealed the 3′ end of the TE sequence with about 30 b.p. polyA tail and polyA site located 29 b.p. downstream of the polyadenilation signal TATAAA located in the L1 element (Figure 2—p(A)2).

To quantitatively estimate the amount of chimeric mRNA and to determine whether the TE sequence exonization is the main cause of the patient’s *GAA* dysfunction, we applied the real-time PCR with primers designed to amplify three targets: the junction of exons 14–15 of the *GAA* mRNA, the junction of the *GAA* exon 15 and the 5′ region of TE sequence and the 3′ end of TE sequence. The corresponding amplicons were cloned into plasmid vector, which was used as a reference sample for ddCq calculation. The relative amount of the exon 15-TE 5′ and TE 3′ molecules normalized to the expression of exons 14–15 was 0.22 and 0.58, respectively. In addition, no significant amplification of exon 15-TE 5′ and TE 3′ target was detected in five control samples.

The results of the allelic imbalance analysis for the c.642C>T and c.1374C>T variants in the mother’s cDNA demonstrated that the TE-containing isoform is reduced to about 30% of the WT isoform (Figure 1D). So, if we consider the allelic imbalance analysis data, the amount of the TE-containing isoform should be about 25% (66.6% (total *GAA* mRNA level—50% + 1/3*50%) / 16.6%) of total *GAA* isoforms in the mother’s cDNA. This is in a good agreement with the data from chimeric isoform measurement for ex 15-TE 5′—22%. The most probable reason for high TE 3′ expression (58%) is that the cDNA was synthesized using oligo(dT) primers, which cause the 3′ end bias, especially in the chimeric isoform, containing about 2000 b.p. of the GC-rich sequence between the TE 3′ target and the reference ex 14–15 target. The 3′ end bias is also the reason for using primers for the *GAA* exons 14–15 instead of exons 2–3 for the chimeric isoform measurement, as they are located as close as possible to the *GAA*-TE junction.

Together, these results strongly suggest that the exonization of the ~3137 b.p. TE sequence using strong acceptor site right downstream of CCCTCT repeats and polyA signal in the 3′ end of the TE with subsequent termination of transcription is the main deleterious consequence of this insertion.

## 4. Discussion

The majority of the human genome is occupied by TEs [21,22]. TE-derived sequences provide to the genomes the large number of functional and regulatory elements, including promoters, enhancers, transcription terminators, small RNA genes and also contribute to the chromatin structure [23,24,25,26,27,28]. 

Most of the TEs in differentiated somatic cells are unable to transpose, being silenced by heterochromatization, DNA methylation or RNA-interference [29,30]. Several classes of TE, including *Alu*, L1 and SVA, could be activated and mobilized by the transposition machinery encoded by autonomous TEs. After integration into the genomic loci TEs can alter the genes expression and contribute both to the complex common diseases being TE polymorphisms and to monogenic diseases [14,31,32]. A vast majority of TE insertions associated with human monogenic diseases are *Alu* elements, which integrate into exons and disrupt the gene’s coding sequence [14].

One of the youngest classes of TEs is SVA. SVAs (SINE-VNTR-*Alu*) are nonautonomous hominid-specific retrotransposable elements that are highly polymorphic and active in the human population [33,34]. Typical SVA is 2 kb in length and consists of CCCTCT repeats, which could serve as an internal promoter for PolII, two *Alu*-like sequences in antisense orientation, GC-rich variable number tandem repeats (VNTR) and short interspersed nuclear elements of retroviral origin (SINE-R). Most SVAs are full length but are known to transduce 5′ and 3′ flanking sequences during transposition [34,35,36]. SVA can impact the genomic loci around its insertion site at DNA, RNA or epigenetic level and alter the nearby genes’ expression [37].

More than 13 SVA insertions are reported as a cause of human disease [14]. The mechanisms of their pathogenesis include alteration of gene’s splicing through the exonisation of TE sequences, the alteration of methylation patterns, transposition-associated deletions and the disruption of the gene’s coding sequence.

The homozygous 3394 b.p. insertion identified in the *GAA* intron 15 represents the complex TE, composed of SVA_D, SVA_E elements in forward orientation and the L1ME3 element in the reverse orientation. The TE insertion carries the typical footprints of a retrotransposition event, which are several insertions of the guanine nucleotide at the 5′ end (at CCCTCT repeat) formed during the mRNA capping, the ~117 b.p. polyA tail at the 3′ end of L1ME3, the 15 b.p. target site duplication formed during target-primed reverse transcription of TE RNA and the specific integration site sequence in the *GAA* intron 15. This strongly suggests that the insertion is formed after the activation and transcription of the polymorphic CPX-1 transposon located at chr20, the transduction of L1ME3 element at which the transcription termination occurred and the subsequent reverse transcription and integration of TE in the *GAA* intron 15.

The detailed functional analysis revealed the underlying mechanism of pathogenesis, which leads to the almost complete absence (~1%) of the full-length *GAA* mRNA isoform and represents the exonization of the ~3137 b.p. TE sequence and the termination of transcription after one of polyA signals located in it. The transcription termination by the TE sequences located in genes’ bodies is a widespread mechanism, which is associated with reduced gene expression and proposed to be one of the reasons for negative evolutionary selection against sense-oriented TE insertions [38,39]. It is estimated that 28% of human gene loci have at least one TE-derived transcription termination signal, among which 70% lead to the synthesis of truncated transcripts [24]. In our case, the L1ME3 element, located in the antisense orientation in relation to the preceding SVA elements and to the *GAA* gene, effectively terminated the transcription both at the stage of TE RNA transcription at chromosome 20 and during the *GAA* pre-mRNA processing after integration of TE into the *GAA* locus. The results of the qPCR and allelic imbalance analyses demonstrated an almost complete absence of the wild type transcript isoform and the presence of truncated isoforms lacking exons 16 to 20 of *GAA* (223 out of 952 amino acids of GAA), which most likely leads to the synthesis of non-functional proteins. Although the activity of the cryptic splice site located in TE and premature transcription termination could be tissue-specific and lead to the synthesis of various amount of wild-type transcript in tissues other than blood, the patient’s severe phenotype and the family history strongly suggests that this insertion is a severe mutation, which is associated with infantile-onset form of PD.

## 5. Conclusions

The results of our study revealed the novel type of mutation, associated with infantile-onset PD. The whole genome sequencing followed by comprehensive functional analysis allowed us to discover and characterize the deleterious effect of the complex mobile genetic element insertion deep in the intron 15 of the *GAA* gene. This study draws the attention of researchers to the need for detailed analysis of genome sequencing data for the presence of footprints of TE insertions. Currently there is a wide range of freely available bioinformatics algorithms that should be implemented into standard variant calling protocols to detect this underestimated type of mutations. The functional genomics in turn have a wide range of available and effective methods, which could be further used to establish their pathogenicity.

## Figures and Tables

**Figure 1 ijms-22-10887-f001:**
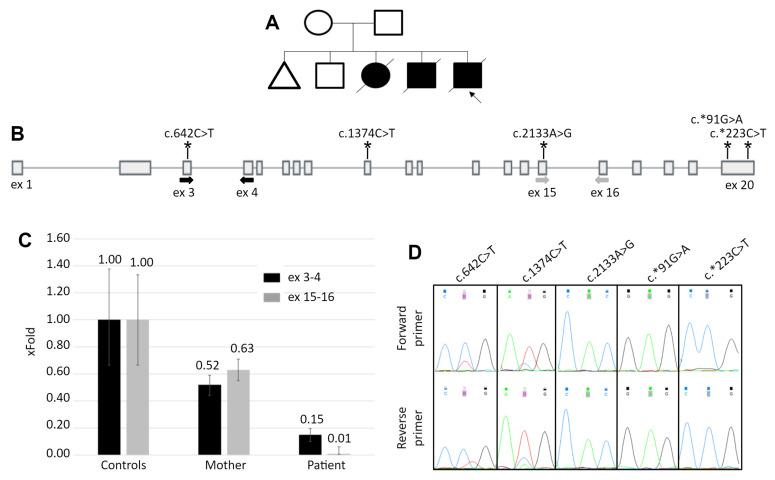
The patient’s pedigree and results of the white blood cells mRNA analysis. (**A**) The patient’s pedigree. Blacked out symbol—patient with the PD phenotype; crossed out—deceased patient; triangle—miscarriage. (**B**) The scheme of the *GAA* gene. Bold arrows indicate location of the primers for qPCR and asterisks indicate the location of heterozygous variants, identified in the mother’s DNA. (**C**) The results of qPCR for the *GAA* cDNA with primers spanning exons 3–4 and 15–16. Controls—mean and standard deviation for 5 healthy control samples. For the mother’s and the patient’s samples the standard deviation was calculated from technical replicates. (**D**) Sanger chromatograms of the mother’s cDNA fragments containing heterozygous variants. The allelic imbalance demonstrates the similar to the qPCR data, expression pattern.

**Figure 2 ijms-22-10887-f002:**
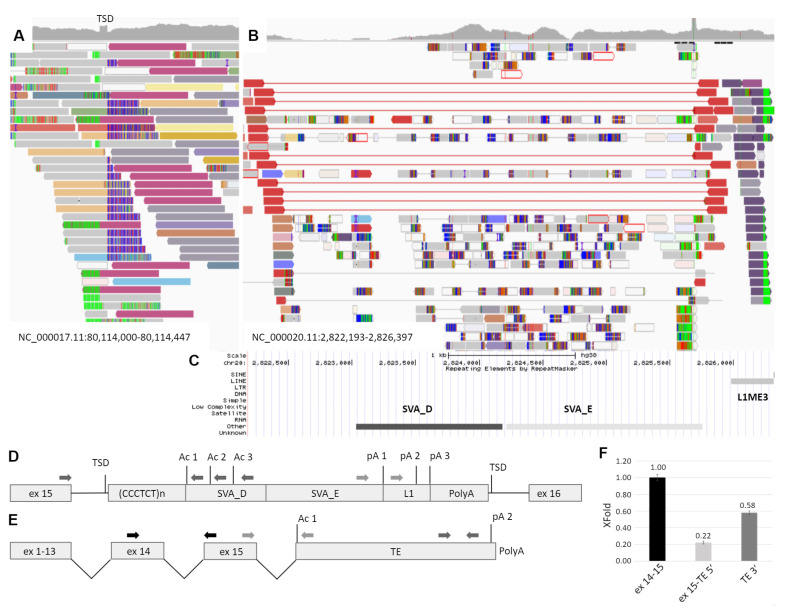
Identification and characterization of the TE insertion. (**A**) The IGV browser window representing the discordant and split reads mapped to the intron 15 of the *GAA* gene. TSD—target site duplication formed during TE integration and represented by the increased coverage. (**B**) Reads mapped to the region of the transposition origin at chr20. The red reads flank the polymorphic CPX-1 transposon presented in patient in heterozygous state. Dark blue reads are discordant, and their pairs are mapped to the *GAA* intron 15 (dark red reads at Figure 1A). (**C**) The RepeatMasker track of the UCSC browser aligned with IGV and representing the mobile genetic elements in the origin of transposition. (**D**) The scheme of TE insertion in *GAA* intron 15 at the DNA level. Ac 1–3—several strong acceptor splice sites; pA 1–3—several transcription termination signals. (**E**) The scheme of the chimeric *GAA* transcript isoform with exonization of TE. (**F**) The results of transcript-specific qPCR of the mother’s cDNA sample. The relative amounts of different parts of the chimeric *GAA* transcript isoform are presented.

## Data Availability

The data was deposited in the ClinVar database (Variation ID: 1240407).

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
