# Peer review of "Complex Transposon Insertion as a Novel Cause of Pompe Disease"

_ijms, 2021, doi:10.3390/ijms221910887_

Round 1

Reviewer 1 Report

In this paper the authors describe a novel molecular genetic cause of Pompe Disease. The approach used to analyze this case, in my opinion, sounds good. I suggest some minor revision before to accept it for publication.

Please specify the total length and genome location of GAA gene and the number of amino acids of the protein encoded.

Page4: the authors write: “Assuming that the reduced activity of the GAA promoter could affect the expression of EIF4A3, we measured its expression and revealed that it did not differ from control.” How do they measure this activity? Explain better this sentence or eliminate it.

Figure 2A. this figure is not clear for me. They say: “The red reads flank the polymorphic CPX-1 transposon presented in patient in heterozygous state. Dark blue reads flank the boundaries of the transposed sequence, and their pairs are mapped at the GAA intron 15.” I consider this caption a little bit confused. Which part of the figure represent the CPX-1 transposon?  

Discussion: please cite a recent review in which are described all the possible role of TE as regulatory elements in cis. (doi:10.3390/biology9020025)

Author Response

Dear reviewer, we thank you for your comments. We highlighted our answers in bold.

Reviewer 1.

In this paper the authors describe a novel molecular genetic cause of Pompe Disease. The approach used to analyze this case, in my opinion, sounds good. I suggest some minor revision before to accept it for publication.

Please specify the total length and genome location of GAA gene and the number of amino acids of the protein encoded.

Added to the introduction para 3.

Page4: the authors write: “Assuming that the reduced activity of the GAA promoter could affect the expression of EIF4A3, we measured its expression and revealed that it did not differ from control.” How do they measure this activity? Explain better this sentence or eliminate it.

We measured it by qPCR with primers to EIF4A3 gene and housekeeping genes. The sequences of primers are in the supplementary file. We added “by qPCR” to this sentence.

Figure 2A. this figure is not clear for me. They say: “The red reads flank the polymorphic CPX-1 transposon presented in patient in heterozygous state. Dark blue reads flank the boundaries of the transposed sequence, and their pairs are mapped at the GAA intron 15.” I consider this caption a little bit confused. Which part of the figure represent the CPX-1 transposon?  

We made the more demonstrative figure and legend.

Discussion: please cite a recent review in which are described all the possible role of TE as regulatory elements in cis. (doi:10.3390/biology9020025).

Sited in discussion para 1.

Reviewer 2 Report

The authors describe novel transposable element (TE) in intron 15 of the GAA gene of a Pompe disease patient (homozygous TE, with confirmed strong reduction of GAA activity) and his mother (heterozygous TE). In the mother, the GAA transcript is reduced by 50%, and in the patient even stronger. Exons downstream of the TE are not expressed neither in the patient nor the mother. The work of the authors describes well their way to identifying this unknown alteration in the GAA gene. Through whole genome sequencing the authors were able to define the TE and determine its origin in chromosome 20.

Major concerns:

  1. Though the authors find arguments for how the TE could lead to decreased GAA mRNA levels, they fail to provide final proof. In the mother, the relative expression of the TE-containing exon was about half of that of exons 14-15, which represents both mRNA copies: with and without TE. This means that TE-mRNA is about half of the total GAA mRNA in the heterozygous mother. This would imply that the TE does not have an impact on mRNA stability. However, this finding is inconsistent with the very strong downregulation of all GAA transcript in the homozygous patient. The authors do not adequately discuss these discrepancies and do not provide proof that the new TE is causing the GAA mRNA reduction.
  2. I think the introduction would benefit from including more detail on the GAA gene and transposable elements. E.g. How many exons? Are mutations clustered in certain exons or introns, or are they evenly spread out across the whole gene? Also, are there any known correlations between mutation locus and enzymatic activity? Give a brief introduction of transposable elements and their relevance to human health. IN the results part the authors state "As the GAA intron 15 already contains TE of various types". This should be introduced in the introduction. What TEs in intron 15 of GAA are known? What is there established effect? (some of that comes in the discussion, but it would serve the article to bring that on earlier)
  3. The authors write 'Primers were designed to provide almost 100% reaction efficiency and high specificity'. Please describe how primer efficiency and specificity was confirmed. The authors compare results from different qPCR assays, efficiency analyses are necessary to make comparisons. Please show data on how primer efficiencies were determined.
  4. The authors state "The results of qPCR revealed that the patient's GAA expression is reduced to 15% of the control at 5′ region and to 1% at 3′ region (Fig. 1 C)." Please show how 5' and 3' expression differs in control samples. If both qPCRs are close to 100% efficient, 5' and 3' expression should be virtually identical.
  5. How does expression of exon3-4, exon 15-16, and exon 14-15 in the mother compare? Why did the authors not use exon3-4 as a reference in Figure 2E?
  6. The authors use words such as comprehensive or detailed when they describe the functional characterization of the TE. This should be toned down given that the authors did not provide final proof that the TE causes the reduced expression and eventually less GAA activity. This would require reconstitution of the TE-containing transcript e.g. in cell culture and protein expression with comparison of the truncated version to WT GAA, which the authors did not do.

Minor concerns:

  1. 1A: please describe colors and symbols in the pedigree.
  2. 1C: Please explain where control samples were obtained and what the error bars represent. The variation in the control bars seems to be quite high. How do the authors explain this?
  3. 1D: please use lines or white space to separate chromatogram regions that are not continuous. The way it is presented, almost seems like one continuous chromatogram. The individual pieces should be visible. Also, please increase the font size denoting the bases above the chromatogram. They are hardly legible.
  4. 2C: please explain the annotations TSD, Ac1, Ac2, Ac3, pA1, pA2, pA3 in the figure legend.
  5. Some language issue (not complete):
    1. abstract: should be "none of which ARE associated with..."
    2. Intro: "PD is caused by a defective activity of.." should be "PD is caused by a deficiency of... ". And it should go on "acid alpha-glucosidase (GAA), a lysosomal enzyme"
    3. intro: put a comma in "oxidative stress, and mitochondrial abnormalities"
    4. intro: add AND in "cardiac, and smooth"
    5. intro: 'analysis of the GAA' you could add the word 'gene' to make it clearer.
    6. Results: should be 'The patient is a 1,5 month old male from the fifth birth in a consanguineous marriage.'
    7. Results: no THE in 'Based on the anamnesis and the family history, PD was suspected.'
    8. Results: should be 'At the next step, whole genome sequencing was performed using the patient’s DNA.'
    9. results: better 'mapping at a chromosome 20 region that contains a polymorphic SVA element named CPX-1'
    10. general: do not use THE before Sanger sequencing.
    11. results: "is the methylation of THE corresponding locus "
    12. results: better is "we hypothesize, THAT the TE sequence could be spliced...". I wonder whether the authors mean this: "we hypothesize, that GAA mRNA splicing could render the TE sequence the last exon terminating the transcription."
    13. results: add THE in "we performed PCR using THE mother's cDNA..."
    14. discussion: better is "The homozygous 3394 b.p. insertion identified in the GAA intron 15 represents a complex TE, composed of SVA_D, SVA_E elements in forward orientation and L1ME3 element in the reverse orientation. "
    15. discussion: better is "attention of researcheRs to the need "

Author Response

Dear reviewer, we thank you for your comments. We highlighted our answers in bold.

Reviewer 2.

The authors describe novel transposable element (TE) in intron 15 of the GAA gene of a Pompe disease patient (homozygous TE, with confirmed strong reduction of GAA activity) and his mother (heterozygous TE). In the mother, the GAA transcript is reduced by 50%, and in the patient even stronger. Exons downstream of the TE are not expressed neither in the patient nor the mother. The work of the authors describes well their way to identifying this unknown alteration in the GAA gene. Through whole genome sequencing the authors were able to define the TE and determine its origin in chromosome 20.

Major concerns:

  1. Though the authors find arguments for how the TE could lead to decreased GAA mRNA levels, they fail to provide final proof. In the mother, the relative expression of the TE-containing exon was about half of that of exons 14-15, which represents both mRNA copies: with and without TE. This means that TE-mRNA is about half of the total GAA mRNA in the heterozygous mother. This would imply that the TE does not have an impact on mRNA stability. However, this finding is inconsistent with the very strong downregulation of all GAA transcript in the homozygous patient. The authors do not adequately discuss these discrepancies and do not provide proof that the new TE is causing the GAA mRNA reduction.

We reanalyzed the data and found a mistake, that lead to the incorrect amount of TE-containing isoform. The correct value for ex 15-TE 5` expression (Figure 2F) is 0,22. The amount of TE 3` expression remains 0,58. The reason for it is that the cDNA was synthesized using oligo(dT) primers which cause the 3` end bias. The patient`s mother lives in the other country and we cannot obtain her blood for cDNA synthesis with R6 primers, which is more appropriate for qPCR. The 3` end bias is also the reason for using primers for the GAA exons 14-15 instead of 2-3 for chimeric isoform measurement (it is also the answer to the Reviewer`s comment 5). Initially, we designed the plasmid containing exons 2-3, exon 15-TE 5` and exons 15-16 and find out that 3` end bias leads to almost threefold difference between GAA molecules, containing exons 2-3 and 15-16 in control samples. In the TE-containing mRNA molecule, there is also about 2000 b.p. of GC-rich repeats, which further bias the mothers chimeric cDNA molecules towards the 3` fraction. To minimize this bias, we designed primers for reference target (exons 14-15) as close as possible to GAA – TE junction. This reference target and plasmid DNA molecules, containing all 3 targets in the same amount (as reference sample) were used to measure the relative amount of the chimeric molecules (Reviewer`s comment 5), containing 5` end and 3` end of TE.  TE 3` target was used to ensure, that the TE sequence till the identified polyA site was exonized, so the data for TE 3` target expression is mostly supportive for the RACE findings.

Figure 1D demonstrates the allelic imbalance in mothers cDNA for 2 variants located before the TE insertion (c.642C>T and c.1374C>T). Assuming that both WT GAA mRNA molecules would be expressed equally, the amount of the TE-containing isoform is reduced to about 30% of WT isoform. It means that total GAA mRNA level is 50% + 16.6% = 66.6%. So, if we consider allelic imbalance analysis data, the amount of TE-containing isoform is about 25% (66.6% / 16.6%) of total isoforms in mothers cDNA. This is in a good agreement with data from chimeric isoform measurement for ex 15-TE 5` target (which is unbiased as far as it was possible for our conditions) - 22%.

We added the explanation and discussion at the page 7.

  1. I think the introduction would benefit from including more detail on the GAA gene and transposable elements. E.g. How many exons? Are mutations clustered in certain exons or introns, or are they evenly spread out across the whole gene? Also, are there any known correlations between mutation locus and enzymatic activity? Give a brief introduction of transposable elements and their relevance to human health. IN the results part the authors state "As the GAA intron 15 already contains TE of various types". This should be introduced in the introduction. What TEs in intron 15 of GAA are known? What is there established effect? (some of that comes in the discussion, but it would serve the article to bring that on earlier)

We expanded the introduction, according to this comment. The TE in intron 15 of GAA is uncharacterized as well as more than 10 other TEs located in GAA. They are identified only by RepeatMasker.

  1. The authors write 'Primers were designed to provide almost 100% reaction efficiency and high specificity'. Please describe how primer efficiency and specificity was confirmed. The authors compare results from different qPCR assays, efficiency analyses are necessary to make comparisons. Please show data on how primer efficiencies were determined.

We added the results of the standard curve and melt curve analyses for all of the GAA qPCR primers in the supplementary file. There were two assays – one for blood cDNA analysis and second for chimeric isoform analysis in mother`s cDNA sample. We do not compare them quantitatively, we just say that the results are similar and supportive to each other and to the allelic imbalance analysis results.

  1. The authors state "The results of qPCR revealed that the patient's GAA expression is reduced to 15% of the control at 5′ region and to 1% at 3′ region (Fig. 1 C)." Please show how 5' and 3' expression differs in control samples. If both qPCRs are close to 100% efficient, 5' and 3' expression should be virtually identical.

We answered in the first comment.

  1. How does expression of exon3-4, exon 15-16, and exon 14-15 in the mother compare? Why did the authors not use exon3-4 as a reference in Figure 2E?

We answered in the first comment.

  1. The authors use words such as comprehensive or detailed when they describe the functional characterization of the TE. This should be toned down given that the authors did not provide final proof that the TE causes the reduced expression and eventually less GAA activity. This would require reconstitution of the TE-containing transcript e.g. in cell culture and protein expression with comparison of the truncated version to WT GAA, which the authors did not do.

This data do not provide proof of how the identified TE is causing the GAA mRNA reduction, because the reduction of chimeric GAA-TE isoform is probably due to reduced processing near 3` end and stability - the properties that we did not assess. But, what is more important, the joint data strongly suggest that the major molecular genetic mechanism of the GAA deficiency in our patient is the premature transcription termination in the vast majority of GAA mRNA molecules and synthesis of truncated protein which lacks 5 exons which is about ¼ of GAA aminoacids. There are several functionally characterized pathogenic splicing variants, which lead to skipping of  these exons and cause the GAA deficiency. So we think that such large deletion is most likely deleterious, lead to complete absence of GAA activity and severe infantile phenotype.

Minor concerns:

  1. 1A: please describe colors and symbols in the pedigree.

Done.

  1. 1C: Please explain where control samples were obtained and what the error bars represent. The variation in the control bars seems to be quite high. How do the authors explain this?

We added the information to the Figure 1 legend. All cDNA samples were synthesized from white blood cells RNA. The error bars represent the standard deviation between 5 control samples. Given the fact, that the GAA gene is highly polymorphic (and many polymorphisms are located in the promoter region, splice sites and UTRs), SD±0,4 is quite normal in our opinion.

  1. 1D: please use lines or white space to separate chromatogram regions that are not continuous. The way it is presented, almost seems like one continuous chromatogram. The individual pieces should be visible. Also, please increase the font size denoting the bases above the chromatogram. They are hardly legible.

Done.

  1. 2C: please explain the annotations TSD, Ac1, Ac2, Ac3, pA1, pA2, pA3 in the figure legend.

We added the information to the Figure 2 legend.

  1. Some language issue (not complete):
    1. abstract: should be "none of which ARE associated with..."
    2. Intro: "PD is caused by a defective activity of.." should be "PD is caused by a deficiency of... ". And it should go on "acid alpha-glucosidase (GAA), a lysosomal enzyme"
    3. intro: put a comma in "oxidative stress, and mitochondrial abnormalities"
    4. intro: add AND in "cardiac, and smooth"
    5. intro: 'analysis of the GAA'you could add the word 'gene' to make it clearer.
    6. Results: should be 'The patient is a 1,5 month old male from the fifth birth in a consanguineous marriage.'
    7. Results: no THE in 'Based on the anamnesis and the family history, PD was suspected.'
    8. Results: should be 'At the next step, whole genome sequencing was performed using the patient’s DNA.'
    9. results: better 'mapping at a chromosome 20 region that contains a polymorphic SVA element named CPX-1'
    10. general: do not use THE before Sanger sequencing.
    11. results: "is the methylation of THE corresponding locus "
    12. results: better is "we hypothesize, THAT the TE sequence could be spliced...". I wonder whether the authors mean this: "we hypothesize, that GAA mRNA splicing could render the TE sequence the last exon terminating the transcription."

We mean that the TE sequence render the GAA mRNA the new last exon terminating the transcription, as it was observed.

  1. results: add THE in "we performed PCR using THE mother's cDNA..."
  2. discussion: better is "The homozygous 3394 b.p. insertion identified in the GAA intron 15 represents a complex TE, composed of SVA_D, SVA_E elements in forward orientation and L1ME3 element in the reverse orientation. "
  3. discussion: better is "attention of researcheRs to the need "

All of the corrections were made.

Round 2

Reviewer 2 Report

The authors describe novel transposable element (TE) in intron 15 of the GAA gene of a Pompe disease patient (homozygous TE, with confirmed strong reduction of GAA activity) and his mother (heterozygous TE). In the mother, the GAA transcript is reduced by 50%, and in the patient even stronger. Exons downstream of the TE are not expressed neither in the patient nor the mother. The work of the authors describes well their way to identifying this unknown alteration in the GAA gene. Through whole genome sequencing the authors were able to define the TE and determine its origin in chromosome 20. In this revision, after recalculating part of their data, the authors find evidence that the TE causes the pathogenic decrease in GAA mRNA levels. The authors greatly improved their manuscript in this revision and provided comprehensive answers to the reviewer’s concerns.

Major concerns:

None.

Minor concerns:

  1. 1 legend. The last sentence “The allelic imbalance demonstrates the similar to the qPCR data, expression pattern.” is unclear. Please explain in the results text how the allelic imbalance is reflecting the qPCR results.
  2. 2 legend: Panel notation is incorrect. "D" should now be E, and "E" should now be F.
  3. Some language issue (not complete):
    1. Abstract: better "we report a novel molecular genetic cause"
    2. Abstract: better "to discover and characterize a complex mobile genetic..."

Author Response

Major concerns:

None.

Minor concerns:

  1. 1 legend. The last sentence “The allelic imbalance demonstrates the similar to the qPCR data, expression pattern.” is unclear. Please explain in the results text how the allelic imbalance is reflecting the qPCR results.

We explained it in the Page 5, Para 3 – “Sequencing of the cDNA fragments containing these variants demonstrated the characteristic allelic imbalance, similar to the expression pattern in patient’s cDNA – partial loss of zygosity for two variants in the 5′ half of the gene and complete loss of zygosity for three variants in the 3′ half (Fig. 1 D).”

  1. 2 legend: Panel notation is incorrect. "D" should now be E, and "E" should now be F.

Corrected

  1. Some language issue (not complete):
    1. Abstract: better "we report a novel molecular genetic cause"
    2. Abstract: better "to discover and characterize a complex mobile genetic..."

Corrected